# Interaction of biomolecules with anatase, rutile and amorphous TiO$_2$ surfaces: A molecular dynamics study

**Tamás Tarjányi**[1], **Ferenc Bogár**[2], **János Minárovits**[1], **Márió Gajdács**[1], **Zsolt Tóth**[3]*

**1** Department of Oral Biology and Experimental Dental Research, Faculty of Dentistry, University of Szeged, Szeged, Hungary, **2** EKLH-SZTE Biomimetic Systems Research Group, Eötvös Loránd Research Network (ELKH), University of Szeged, Szeged, Hungary, **3** Department of Medical Physics and Medical Informatics, University of Szeged, Szeged, Hungary

* ztoth@physx.u-szeged.hu

## Abstract

The adhesion of biomolecules to dental and orthopedic implants is a fundamental step in the process of osseointegration. Short peptide motifs, such as RGD or KRSR, carried by extracellular matrix proteins or coated onto implant surfaces, accelerate cell adhesion and tissue formation. For this reason, understanding the binding mechanisms of adhesive peptides to oxidized surfaces of titanium implants is of paramount importance. We performed molecular dynamics simulations to compare the adhesion properties of 6 peptides, including the tripeptide RGD, its variants KGD and LGD, as well as the tetrapeptide KRSR, its variant LRSR and its truncated version RSR, on anatase, rutile, and amorphous titanium dioxide (TiO$_2$) surfaces. The migration of these molecules from the water phase to the surface was simulated in an aqueous environment. Based on these simulations, we calculated the residence time of each peptide bound to the three different TiO$_2$ structures. It was found that the presence of an N-terminal lysine or arginine amino acid residue resulted in more efficient surface binding. A pulling simulation was performed to detach the adhered molecules. The maximum pulling force and the binding energy were determined from the results of these simulations. The tri- and tetrapeptides had slightly greater adhesion affinity to the amorphous and anatase structure than to rutile in general, however specific surface and peptide binding characters could be detected. The binding energies obtained from our simulations allowed us to rank the adhesion strengths of the studied peptides.

## 1. Introduction

Titanium (Ti) and its alloys have excellent mechanical properties, whereas its surface is always covered by a dense native titanium dioxide (TiO$_2$) layer, providing an extremely good barrier against environmental effects [1, 2]. The outer oxide surface exhibits good biocompatibility with tissues, thus, it is unsurprising that Ti is one of the most commonly used materials in dental and orthopedic implantology [3, 4]. During osseointegration, Ti implants form a connection with the bone, followed by healing and fixation in it. In the early steps of the osseointegration process—during the insertion of the Ti implant—different biomolecules

the study are available from the Zenodo Repository at https://doi.org/10.5281/zenodo.8036651.

**Funding:** This work was supported by the grant GINOP-2.3.2-15-2016-00011, in the initial phase and by the University of Szeged Open Access Fund (Grant number: 6103). The funders had no role in study design, data collection and analysis, decision to publish, or preparation of the manuscript.

**Competing interests:** The authors have declared that no competing interests exist.

from the blood adsorb to the implant surface, establishing an appropriate surface for the subsequent attachment of bone cells [5]. The different biomolecules adsorbed on the TiO$_2$ surface have a great impact on the osseointegration healing process, i.e. the success rate and recovery time [6, 7]. One of the most investigated biomolecules is the three-amino-acid-long peptide RGD (Arg-Gly-Asp) that affects cell attachment. This tripeptide may be found in many extracellular matrix proteins and plays an important role in the osseointegration process [8, 9]. The tetrapeptide KRSR (Lys-Arg-Ser-Arg) has also been suggested as a promising molecule that facilitates osseointegration [9–11]. KRSR was described in different bone adhesive proteins binding to the proteoglycan molecules found on cell surfaces enhancing the adhesive potential of osteoblasts [10, 11].

At room temperature the TiO$_2$ layer, formed on the surface of Ti implants, contains both amorphous TiO$_2$ with disordered structure and stable polymorphic crystalline TiO$_2$ structures, mainly anatase (tetragonal) and rutile (tetragonal) [12, 13]. Based on literature findings, brookite (orthorhombic) a further polymorphic form of TiO$_2$ is said to be very rare on Ti implant surfaces [12, 14, 15]. It is also worth noting that various surface treatments, such as turning or grinding of the metal, or acid treatments considerably affect the structure of the TiO$_2$ layer and its interactions with biomolecules [12, 15]. In this study, we compared the tripeptide RGD, implicated in integrin binding and its variants KGD and LGD, as well as the tetrapeptide KRSR, and its variant LRSR as well as its truncated version RSR, onto the three common TiO$_2$ surface types: anatase, rutile, and amorphous TiO$_2$.

The application of molecular dynamics (MD) simulations has increased considerably in many research fields lately. The MD, based on the laws of classical Newtonian mechanics can be used to perform simulations that elucidate how the different molecular systems behave under the influence of certain pre-determined conditions. Luan et al. previously studied biomolecule adsorption to a TiO$_2$ nanosphere particle in an aqueous environment with MD simulation [16]. In their study, force field parameters were assigned to TiO$_2$, which was a fit to the Buckingham potential of TiO$_2$. Several MD studies were performed on interactions of RGD peptide with TiO$_2$ surfaces. For example, Wu et al. carried out an MD simulation with RGD adsorption on rutile surfaces [17]. It was found that the RGD binding to the rutile surface involves the amide groups, and the attachment mode of this peptide was not influenced by the water models applied in their simulations. Zhang et al. performed MD simulation with RGD on rutile and anatase surfaces and performed pair correlation function analysis for different groups of RGD and atoms of the surface [18]. Based on their findings, the adsorption behavior of RGD on different surfaces was quite similar and the interaction between the peptide and the surfaces was mainly the result of hydrogen bonding. Previously, our study dealt with theoretical simulations describing KRSR adsorption on the anatase TiO$_2$ surface [19]. In that study, a 200 ns simulation was performed followed by a replica exchange molecular dynamics (REMD) simulation that was carried out to map the possible conformations of the molecule on the surface. It was found that the N-terminal of the K (Lys) amino acid of the peptide played an important role in the adsorption process [19]. The KRSR peptide bound to the surface in a stable conformation over time. Furthermore, a pulling simulation on the peptide from the TiO$_2$ surface and umbrella sampling has been performed to investigate the binding characteristics. In this present study, we continued the established MD simulations with well-characterized adhesive peptides RGD and KRSR, as well as with a set of variant peptides not yet studied experimentally or theoretically. These variants, KGD, LGD, LRSR, and RSR differ from the adhesive peptides in a single amino acid residue at the N-terminal position. The aim of this study was twofold: firstly to provide a better understanding of how different TiO$_2$ polymorphs influence the adsorption properties of the biomolecules, secondly to investigate the influence of peptide modifications on the adhesion.

## 2. Methods

### 2.1 MD simulation protocol

MD simulations were performed with GROMACS (2019.6 version). GROMACS is a freely-available software package under the GNU Lesser General Public License [20–27]. The biomolecules KGD (Lys-Gly-Asp), KRSR (Lys-Arg-Ser-Arg), LGD (Leu-Gly-Asp), LRSR (Leu-Arg-Ser-Arg), RGD (Arg-Gly-Asp) and RSR (Arg-Ser-Arg) were created with the polypeptide builder module of the Gabedit 2.5.0 software [28]. The CHARMM36 force field [29, 30] was applied to set up all the parameter files of the biomolecules and their surrounding environment. Anatase and rutile TiO$_2$ unit cells were obtained from the American Mineralogist Crystal Structure Database (AMCSD) [31, 32]. The (100) crystal structures were then built up with the Olex2 software [33]. The cell lengths of the anatase and rutile crystal structure were (x = 2.64 nm, y = 4.58 nm, z = 10.26 nm) and (x = 2.11 nm, y = 4.89 nm, z = 10.67 nm), respectively. The final crystal structures were then placed in a simulation box that had the same lengths as the crystal structures.

The amorphous structure was created from the anatase structure with the aid of GROMACS. To create the amorphous TiO$_2$, the simulation box was set to have a periodic boundary. After a short, 2 ns NVT (keeping the number of particles (N), system volume (V) and temperature (T) constant) and 0.25 ns NPT (keeping the number of particles (N), system pressure (P) and temperature (T) constant) equilibration process, an MD was performed where the temperature was raised to 3000 K. At this temperature, the oxygen and Ti atoms of the TiO$_2$ had enough kinetic energy to move freely from the crystal lattice points. Exploiting the properties of the periodic boundary (i.e., the simulation box repeats itself at the boundary), the system could not expand any further and the volume of the system remained constant. After 0.5 ns MD simulation, the temperature was suddenly set back to 310 K, where a short equilibration procedure (2 ns NVT and 0.25 ns NPT) was performed again, followed by another 2.5 ns MD simulation. At this temperature, the atoms randomly froze in a liquid-like structure, resulting in an amorphous solid structure. This amorphous structure was then used in the subsequent analyses for comparison with the anatase and rutile structures, while the biomolecules adsorbed to the surfaces.

The three different TiO$_2$ structures, namely the anatase, rutile, and amorphous TiO$_2$ are presented in Fig 1. The radial distribution function (RDF) of Ti and O atoms in the final amorphous structure is shown in Fig 2. This function shows the atom distribution as a radial distance. Unlike in crystals, the wide peaks show the disordered arrangement of the constituting atoms.

As the MD simulations are mostly used to simulate proteins in an aqueous environment, the parameters of the TiO$_2$ were not included in the force field and had to be added manually. The Lennard-Jones parameters of TiO$_2$ were applied to set up the interactions, similarly to our previous study [19], which was based on the work of Luan et al. [16]. The parameters used were the following: $\varepsilon_{Ti-Ti} = 0.58$ $kcal/mol$, $\varepsilon_{Ti-O} = 0.424$ $kcal/mol$, $\varepsilon_{O-O} = 0.58$ $kcal/mol$, $\sigma_{Ti-Ti} = 0.220$ $nm$, $\sigma_{Ti-O} = 0.272$ $nm$, $\sigma_{O-O} = 0.324$ $nm$ and the partial charges $q_{Ti} = 2.196e$, $q_O = -1.098e$ in the Lennard-Jones and Coulomb potentials:

$$U\left(r_{ij}\right) = \varepsilon_{ij}\left[\left(\frac{\sigma_{ij}}{r_{ij}}\right)^{12} - 2\left(\frac{\sigma_{ij}}{r_{ij}}\right)^{6}\right] + \frac{q_i q_j}{r_{ij}}$$

where $\varepsilon$, $\sigma$, and $q$ denote the potential well depth, collision diameter, and atomic partial charge, respectively.

**Anatase (100)**

**Rutile (100)**

**Amorphous (TiO2)**

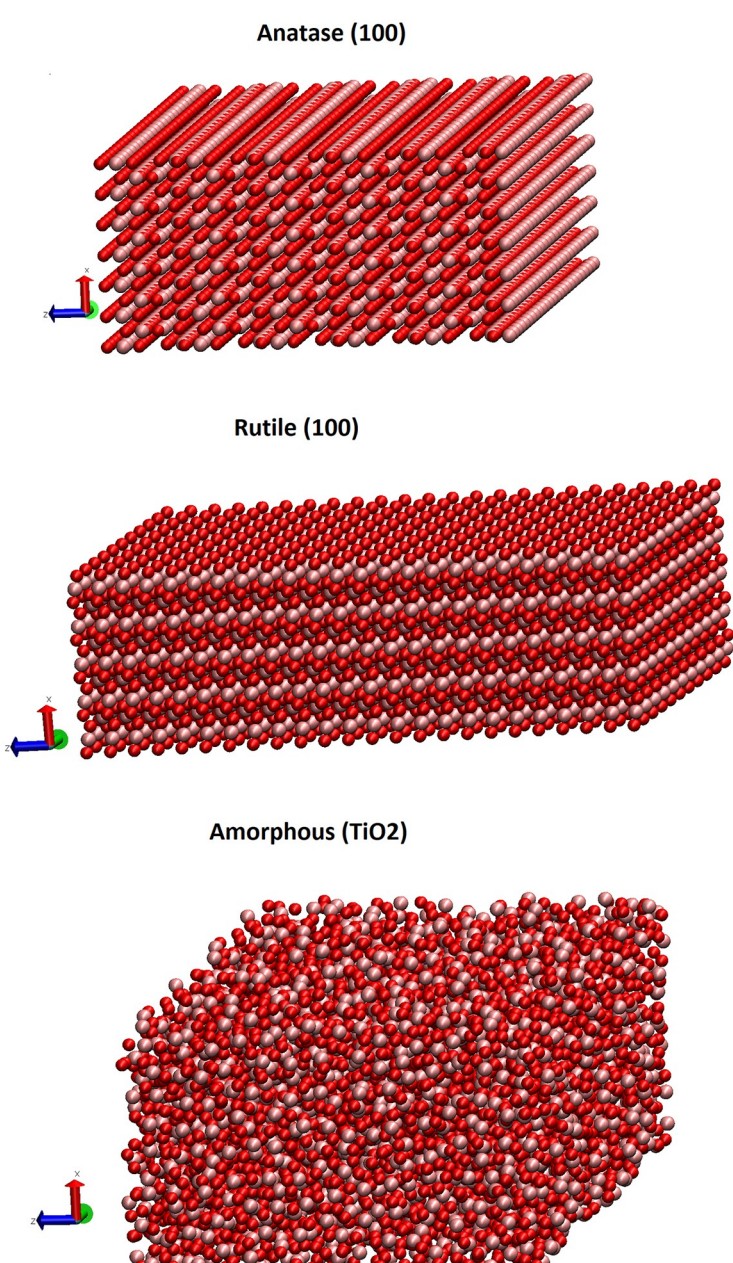

**Fig 1. The geometries of the TiO$_2$ polymorphs that were used for the biomolecular surface adsorption simulations.** Color code: oxygen, red; titanium, pink. Visualization was performed using VMD 1.9.3 program in CPK representation.

After the creation procedure of the TiO$_2$ structures, the simulation box was extended in the x-axis direction with 7 nm giving enough space to add water and the biomolecules. The final length of the simulation box was x = 10 nm. Biomolecules were inserted randomly, near the middle of the newly created empty volume. In total, six different biomolecules were built up, namely KRSR, LRSR, RSR, RGD, KGD, and LGD. The molecular structure of these biomolecules is shown in Fig 3. KRSR and RGD are widely studied peptides, which are involved in the

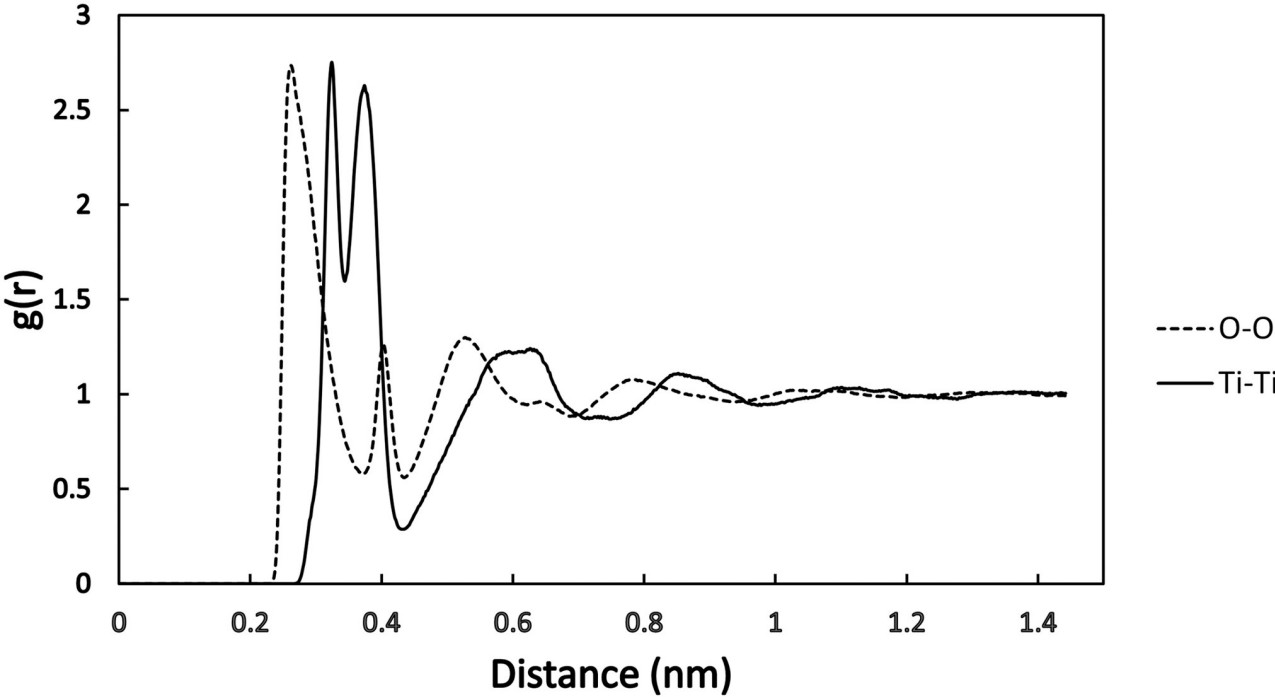

**Fig 2. The radial distribution function (RDF) of oxygen to oxygen and titanium to titanium distances in the amorphous TiO$_2$ structure.**

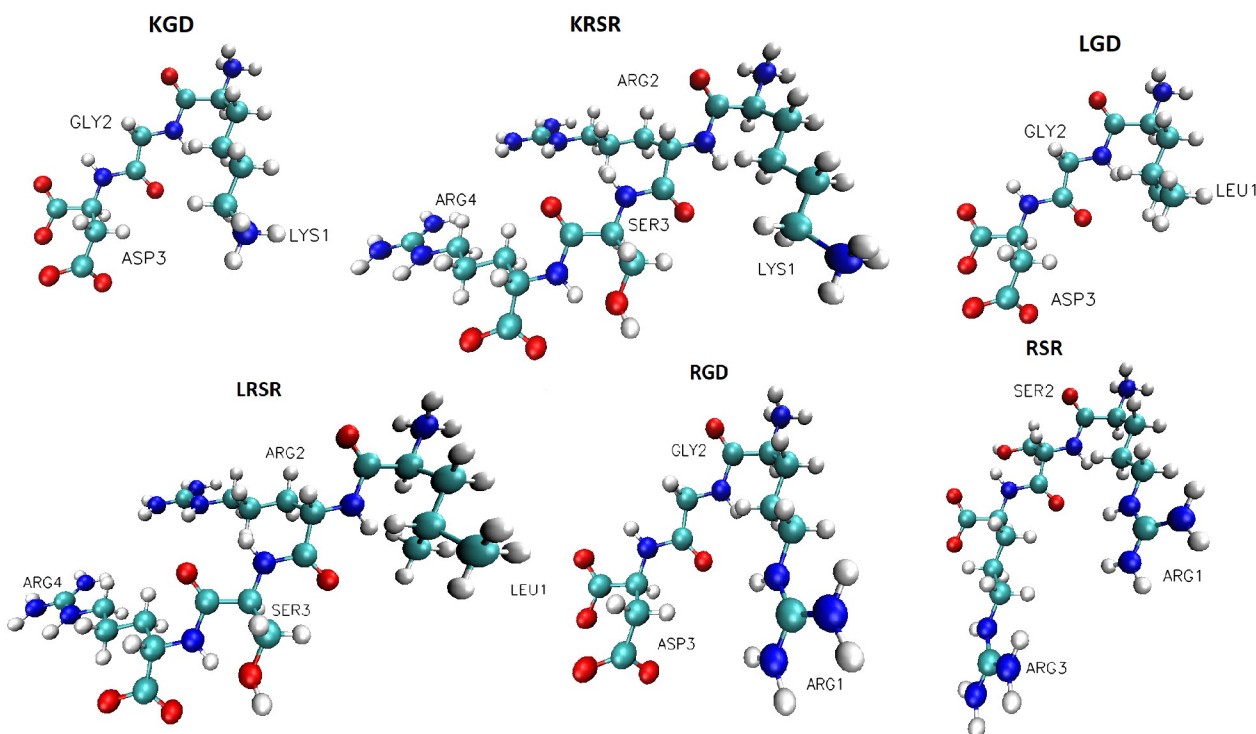

**Fig 3. Biomolecules included in the MD simulations.** Color code: oxygen, red; nitrogen, dark blue; carbon, cyanide blue; hydrogen, white-gray. Visualization was performed using VMD 1.9.3 in CPK representation.

osseointegration processes [5, 8, 9, 10]. With the inclusion of the four additional peptides, we aimed to observe how differently these molecules adsorb to the TiO$_2$ surface, owing to the different binding sites associated with the charged parts of the amino acid molecules. After the abovementioned steps, the solvation of the system was done. The TIP3P explicit water model was used [34], resulting in over 7500 water molecules in the simulation box. The Genion algorithm of GROMACS was used to randomly replace water molecules with Na$^+$ and Cl$^-$ ions. In each case, 5 Na$^+$ and 5 Cl$^-$ ions were added to the system. In addition, extra ions were added in the case of differently charged biomolecules (e.g. 3 Cl$^-$ ions in the case of KRSR) to balance the charge of the overall system.

After the equilibration (NVT and NPT) process, it was noted that some of the water molecules adsorbed to the TiO$_2$ surfaces, resulting in fewer water molecules in the bulk water phase and giving a less dense aqueous environment. To prevent such artifacts during the long MD simulations there was another, second solvation procedure, where water molecules were added once again to the system. A 2000-step energy minimization was performed on each system after each solvation procedure. Before the MD simulations equilibration processes were performed on every system containing a 2.5 ns NVT and 0.5 ns NPT (this was done also before the second solvation and once more thereafter). The temperature and pressure during the equilibration processes were set to 310 K and 1 bar, respectively. The v-rescale thermostat algorithm was used for all the simulations and the temperature was set individually for the groups of water and non-water elements. During all the equilibration processes, there was a position restrain on the backbone of the biomolecules to prevent movements at these steps. The force constant for the position restrain was set to 1000 kJ·mol$^{-1}$·nm$^{-2}$. The time step was set to 2 fs both during the NVT, NPT, and MD simulations. The simulation box boundary was set to fulfill periodic boundary conditions in all cases. The electrostatic and van der Waals cut-offs were set to 1 nm. Finally, a 500 ns MD simulation—using the Verlet algorithm built in GROMACS—was performed for each biomolecule adsorption process to the three different TiO$_2$ structures.

## 2.2 Pulling simulation protocol

The MD simulation characterized the adsorption process of the biomolecules. To calculate the energies, related to adsorption, pulling from the polymorph surfaces was simulated for each adsorbed biomolecule. The conformations of the closest distances of the biomolecules from the surfaces were selected from the 500 ns MD simulations and used as input geometries for the pulling simulation. These distances and the Root Mean Square Deviation (RMSD) of the biomolecules are summarized in the S1 Appendix. The pull simulations were initiated at 310 K with 500 ps NPT equilibration simulation, using 2 fs time steps. During this phase, a 1000 kJ·mol$^{-1}$·nm$^{-2}$ position restrain was set on the biomolecules to prevent them from moving. The electrostatic and van der Waals cut-offs were set up to 1.4 nm. After the NPT, the pull simulations were performed using a harmonic potential having a spring constant set to 1000 kJ·mol$^{-1}$·nm$^{-2}$. All the simulations were prepared to have the x-axis perpendicular to the TiO$_2$ surfaces and the biomolecules were pulled from the surface along this axis. The pulling rate was 10 nm/ns. Data from these pulling simulations were collected, the maximum pulling forces were determined and the binding energies were calculated from the force and distance data with a numerical integration.

## 3. Results

During the MD simulations, it could be seen that the biomolecules (KGD (Lys-Gly-Asp), KRSR (Lys-Arg-Ser-Arg), LGD (Leu-Gly-Asp), LRSR (Leu-Arg-Ser-Arg), RGD (Arg-Gly-

Asp), and RSR (Arg-Ser-Arg)) moved by random Brownian motion in the bulk water phase and approached the TiO$_2$ surfaces. Fig 4a–4f show the distance variation between the biomolecules and the TiO$_2$ surfaces over time. The different biomolecules can adsorb to the surface for different time periods. Adsorption time duration—i.e. the overall time that the molecules spend near the surface—shows considerable variation among the different molecules on different surfaces. The peptides at the closest points adsorbed directly to the surface. However, it can be seen from Fig 4 that the most time was spent by adsorption through the first water layer (i.e. the bonding distance increased approximately with the size of the water molecules, which cover the TiO$_2$ surfaces). The simulation shows that the molecular adsorption changed over time, and after attaching to the surface, the peptides could temporarily move back to the bulk water phase and re-attach again.

Adhesion of the LRSR molecule was the least stable on the different surfaces, the molecule moved back to the water phase frequently (see Fig 4d). Compared to LRSR, the RSR, which does not have the N-terminal L (Leu) amino acid, shows that the adhesion happened on the surfaces for longer periods (Fig 4f). Adding the charged K amino acid, the KRSR had longer, more stable adhesion on the surfaces (Fig 4b). Like KRSR, the RGD tripeptide adsorbs successfully to all the three different types of surfaces, as it is shown in Fig 4e. In this case, the molecule adsorbed very stably in the last 250 ns simulation. Compared to this, in the case of KGD and LGD, the distance between the molecules and the anatase and rutile surfaces changes continuously, while it was stable on the amorphous surface until the end of the simulation time (Fig 4a and 4c).

To compare the adsorption of the different biomolecules on the different surface types the ratio of the adsorbed time to the total simulation time (500 ns) was calculated in each case. For this, an adsorption distance of 0.5 nm was defined, and the adsorption time was calculated as a sum of time durations when the closest atom of the molecules was closer than this adsorption distance. This time ratio was also calculated for the first 250 ns and the second 250 ns of the simulation. The results of the adsorption time ratios are shown in Fig 5a–5c.

The results according to Fig 5a represent an average behavior for the whole simulation time. It clearly shows that the adsorption time ratio of LRSR was the lowest compared to all the other biomolecules included in the study. In Fig 5b and 5c a similar representation can be seen for the first and the second half of the simulation time. Comparing the adsorption time ratios of the second 250 ns-500 ns period to the first 0 ns—250 ns period, it can be observed, that some peptides were stabilized in certain surface types. Total adsorption was reached for KRSR on rutile, LGD, and RSR on amorphous TiO$_2$, while RGD was attached full-time on all surface types.

The MD simulations showed that the molecules could either move freely in the aqueous phase representing an unbound state or located at characteristic distances from the surface during the bound states. In Fig 6 and (Fig A1 in S1 Appendix) the distributions of the peptide-TiO$_2$ surface distances are shown. These relative frequency values were compared to the whole simulation time (all frames of the simulation). The highest relative frequency can be seen in Fig 6a (Fig A1e in S1 Appendix) where RGD was 0.2 nm close to the rutile surface in 64.8% of the whole simulation time. In this case, the RGD is connected with the carboxyl group of the D (Asp.) peptide to the Ti atom in the top layer of the rutile surface. In Fig 6b such molecule conformation is shown at 300 ns simulation time. In this image, the binding occurs without water mediation. In Fig 6c the LGD tripeptide relative frequency distribution is shown. Unlike the RGD, LGD is positioned at further distances (0.3–0.4 nm) from the rutile surface atoms. In the snapshot Fig 6d, the water molecule is apparent between the surface and the carboxyl group of the C-terminal D (Asp) residue of the LGD peptide. Other frequency distributions can be seen in (Fig A1 in S1 Appendix). Higher frequency values occur in the case of KRSR (0.35 nm),

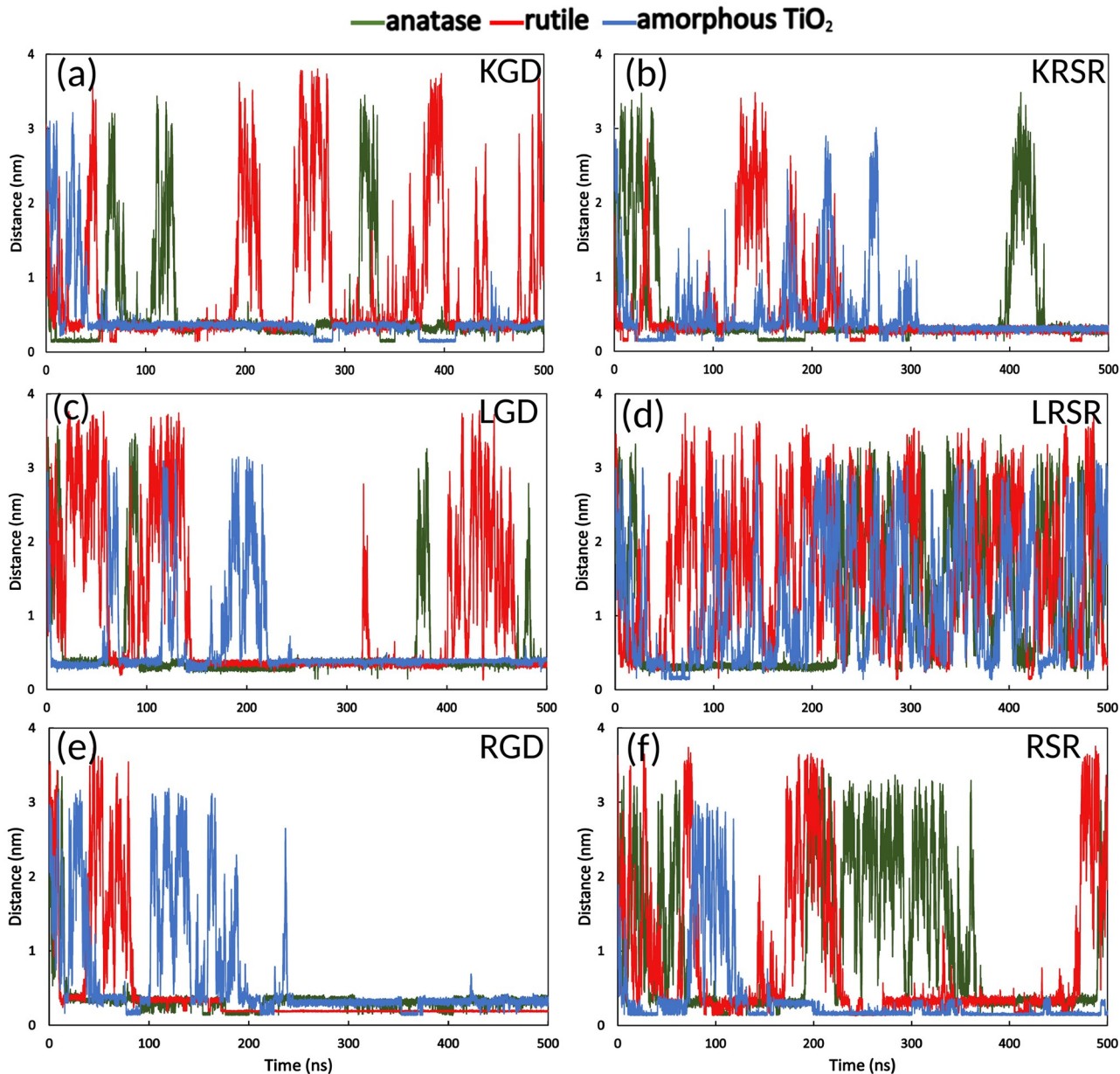

**Fig 4. The distance between the biomolecules and the TiO$_2$ surfaces over time.**

KGD (0.4 nm), and LGD (0.4 nm) on amorphous TiO$_2$ whose values are near 50%. In the case of KRSR on anatase (0.3 nm), the relative frequency is still high 43.84%, while the dominant frequencies of other biomolecules are below 40%.

To compare the nature of the binding of the various peptides to the TiO$_2$ surfaces, pull simulations were performed for each molecule. During the pulling simulations, the center of mass (COM) of the molecules is pulled with the help of a harmonic potential. The change in the position of COM depends on the binding of the molecule to the surface and the interaction with the surrounding molecules. The maximum pulling forces have been determined from the pull simulations, summarized in the Fig A2 in S1 Appendix and Table 1. The smallest maximal

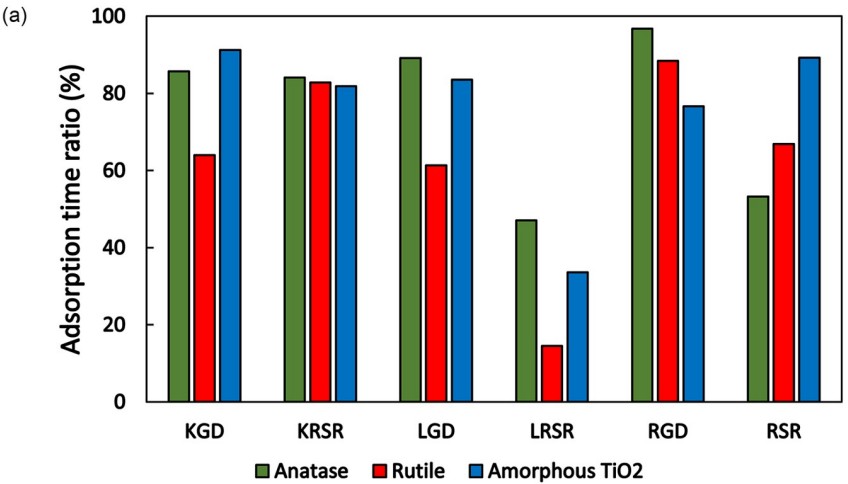

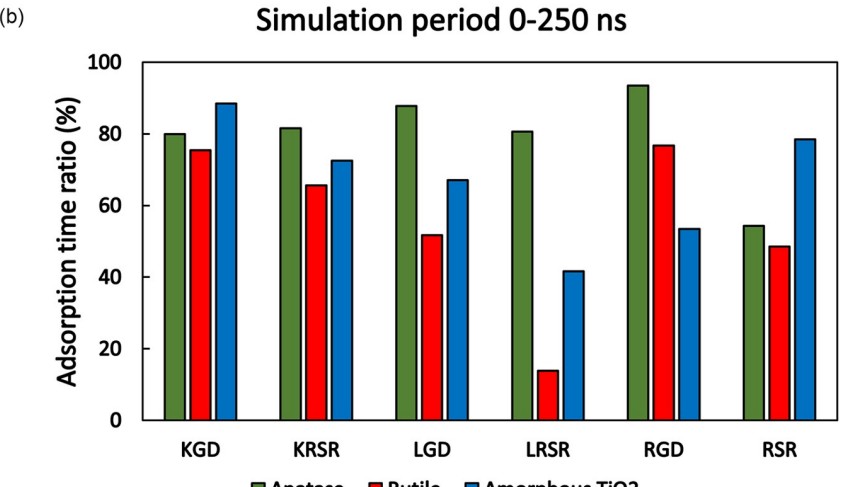

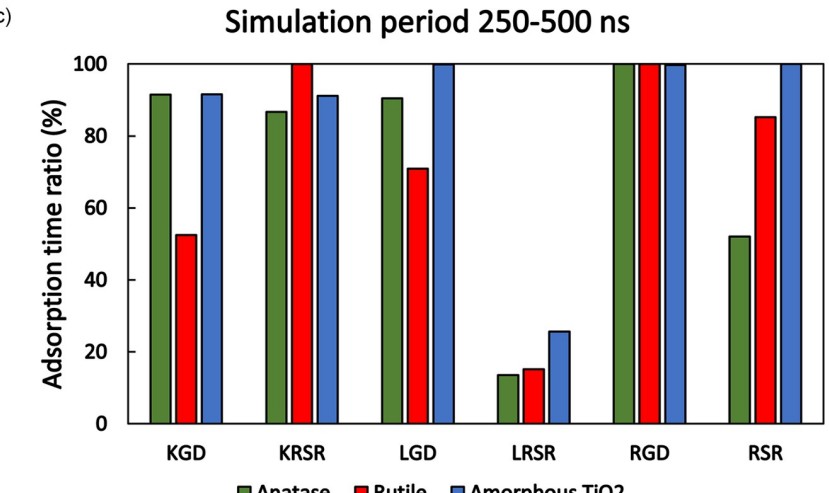

**Fig 5. Adsorption time ratio (a) full simulation time, (b) first 250 ns, and (c) second 250 ns for the studied peptides on the surfaces of the three polymorphic structures of TiO$_2$.** Adsorption distance was defined as 0.5 nm measured from the surface of TiO$_2$.

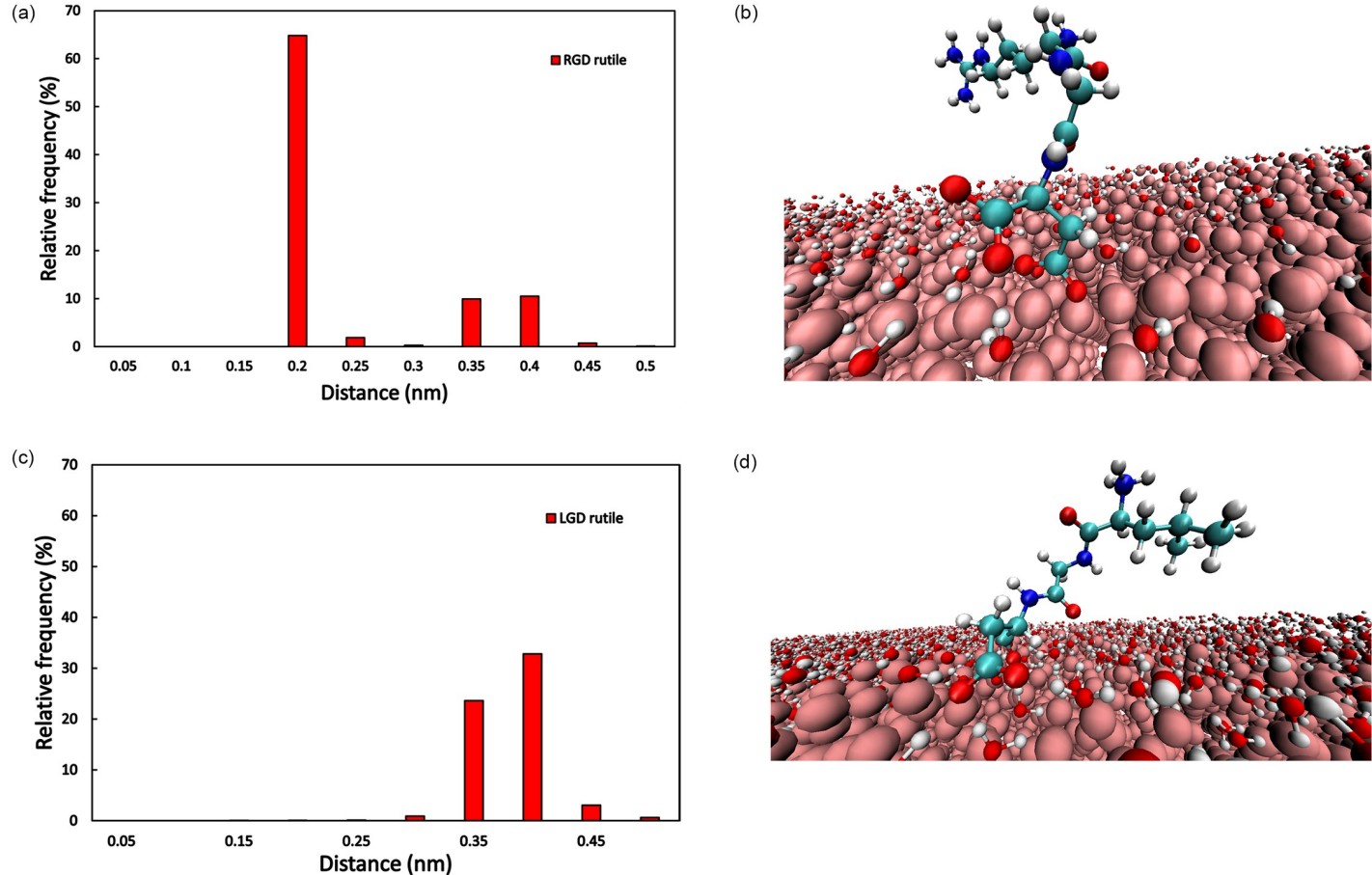

**Fig 6.** Distribution of the (a) RGD and (c) LGD tripeptide-TiO$_2$ surface distances averaged for the whole simulation time. Note, that results are shown up to 0.5 nm only and molecules may spend considerable amount of time also in unbound state, where the distance is larger than this value. Snapshot (b) of RGD bound to Rutile at 300 ns and (d) of LGD bound to rutile at 500 ns. Representation of (b) and (d): peptide CPK, rutile VDW, water molecules CPK, only surface water molecules are shown for clear visualization. Images were made with VMD 1.9.4.

force (238.4 $kJ \cdot mol^{-1} \cdot nm^{-1}$) was found in the case of LRSR on anatase while the largest forces were related to RSR bound to amorphous TiO$_2$ (979.2 $kJ \cdot mol^{-1} \cdot nm^{-1}$), and KRSR on anatase (779.7 $kJ \cdot mol^{-1} \cdot nm^{-1}$). The force value difference between the smallest and highest pulling forces was found to be quite large, 4.1 times greater for RSR on amorphous TiO$_2$ compared to LRSR on anatase.

**Table 1. The maximum values of the pulling forces on the polymorphic TiO$_2$ surfaces of the investigated peptides.** The binding energy was estimated from the force-distance diagram by numerical integration of the curves in the adsorption distance interval.

| Surface | Anatase TiO$_2$ | | | | | | Rutile TiO$_2$ | | | | | | Amorphous TiO$_2$ | | | | | |
|---|---|---|---|---|---|---|---|---|---|---|---|---|---|---|---|---|---|---|
| Peptide | KGD | KRSR | LGD | LRSR | RGD | RSR | KGD | KRSR | LGD | LRSR | RGD | RSR | KGD | KRSR | LGD | LRSR | RGD | RSR |
| Maximum force during pull (($kJ \cdot mol^{-1} \cdot nm^{-1}$)) | 516.3 | 779.7 | 453.0 | 238.4 | 662.7 | 546.1 | 672.9 | 519.3 | 510.3 | 431.6 | 382.0 | 443 | 442.9 | 694.0 | 424.6 | 556.7 | 562.8 | 979.0 |
| Binding energy (kJ/mol) | 335.4 | 690.6 | 276.7 | 124.9 | 559.3 | 346.5 | 415.9 | 330.4 | 329.0 | 165.1 | 238.1 | 255.3 | 225.6 | 419.3 | 182.1 | 395.6 | 337.0 | 754.0 |
| Binding energy (eV) | 3.5 | 7.2 | 2.9 | 1.3 | 5.8 | 3.6 | 4.3 | 3.4 | 3.4 | 1.7 | 2.5 | 2.7 | 2.3 | 4.6 | 1.9 | 4.1 | 3.5 | 7.8 |

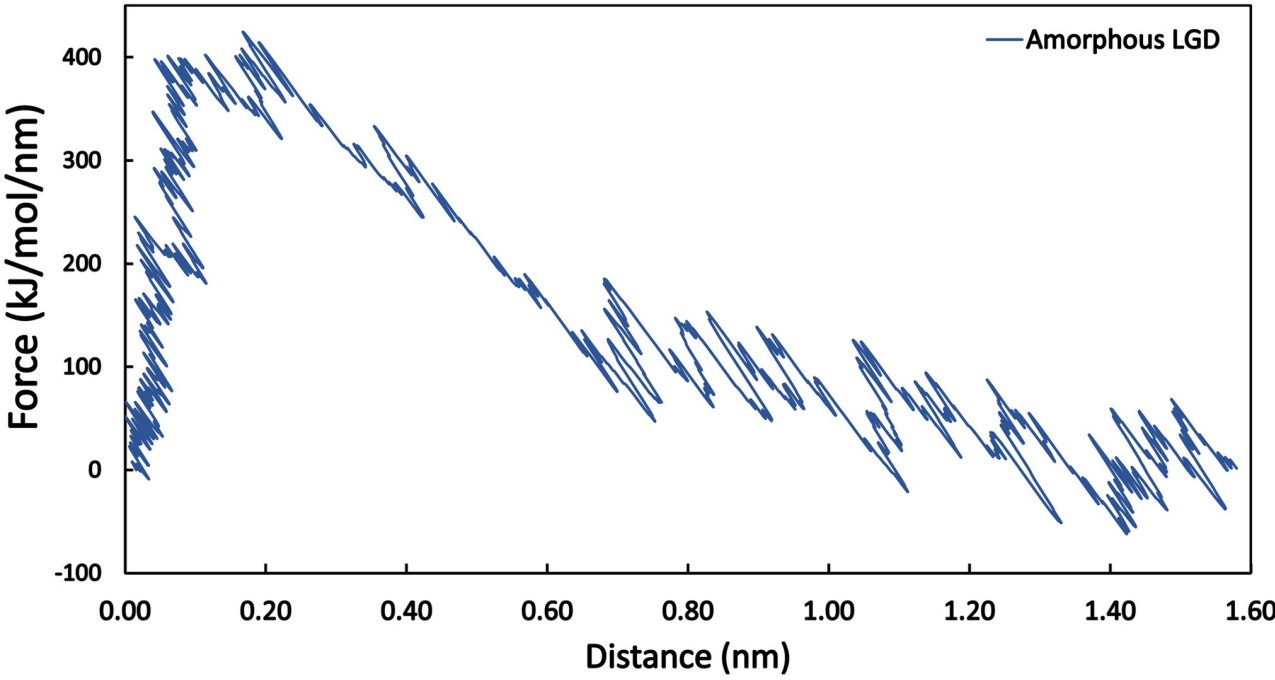

**Fig 7. A representative force-distance diagram showing the pull force as a function of the molecule distance from the TiO₂ surface.**

From the force-distance diagram in Fig 7, it appears that until the maximum force value is reached, the relationship is almost linear between distance and pulling force, which indicates an elastic behavior.

The binding energy was also estimated from the force-distance diagrams with a numerical integration method, which calculates the area under the curves in Fig A2 in S1 Appendix This value can be useful to quantify the degree of adsorption for peptides on different surfaces. Binding energy was smallest in the case of LRSR and anatase (1.29 eV) and for LRSR and rutile (1.71 eV), while the highest binding energies were seen in the case of RSR and amorphous TiO₂ (7.8 eV), and for KRSR and anatase surface (7.15 eV). The maximum pulling force values were also obtained from the results. In Fig 8 the binding energy is plotted as a function of the maximum pulling force. Between these two quantities, a linear correlation is found ($R^2$ = 0.895 and the fitted equation: $E_{binding} = 0.01 \cdot F_{max} - 1.76$).

Mean maximum forces for amorphous TiO₂, anatase and rutile were 610 ± 205.3 $kJ \cdot mol^{-1} \cdot nm^{-1}$, 532 ± 185.1 $kJ \cdot mol^{-1} \cdot nm^{-1}$ and 493 ± 102 $kJ \cdot mol^{-1} \cdot nm^{-1}$, respectively. Mean binding energies for amorphous TiO₂, anatase and rutile were 4.00 ± 2.1 $eV$, 4.03 ± 2.11 $eV$, and 2.99 ± 0.91 $eV$, respectively.

## 4. Discussion

The adhesion processes of tripeptides and tetrapeptides on various crystalline and amorphous TiO₂ surfaces were simulated by our MD calculations. These simulations allowed comparison of adhesion pathways and properties, which may also facilitate cell adhesion and thus influence osseointegration of peptide-coated implants. The 500 ns MD simulations showed that the bio-molecules can successfully bind to the TiO₂ surfaces. This binding occurs mostly through the charged amino acid constituents of the peptides, for example, through the carboxyl groups of

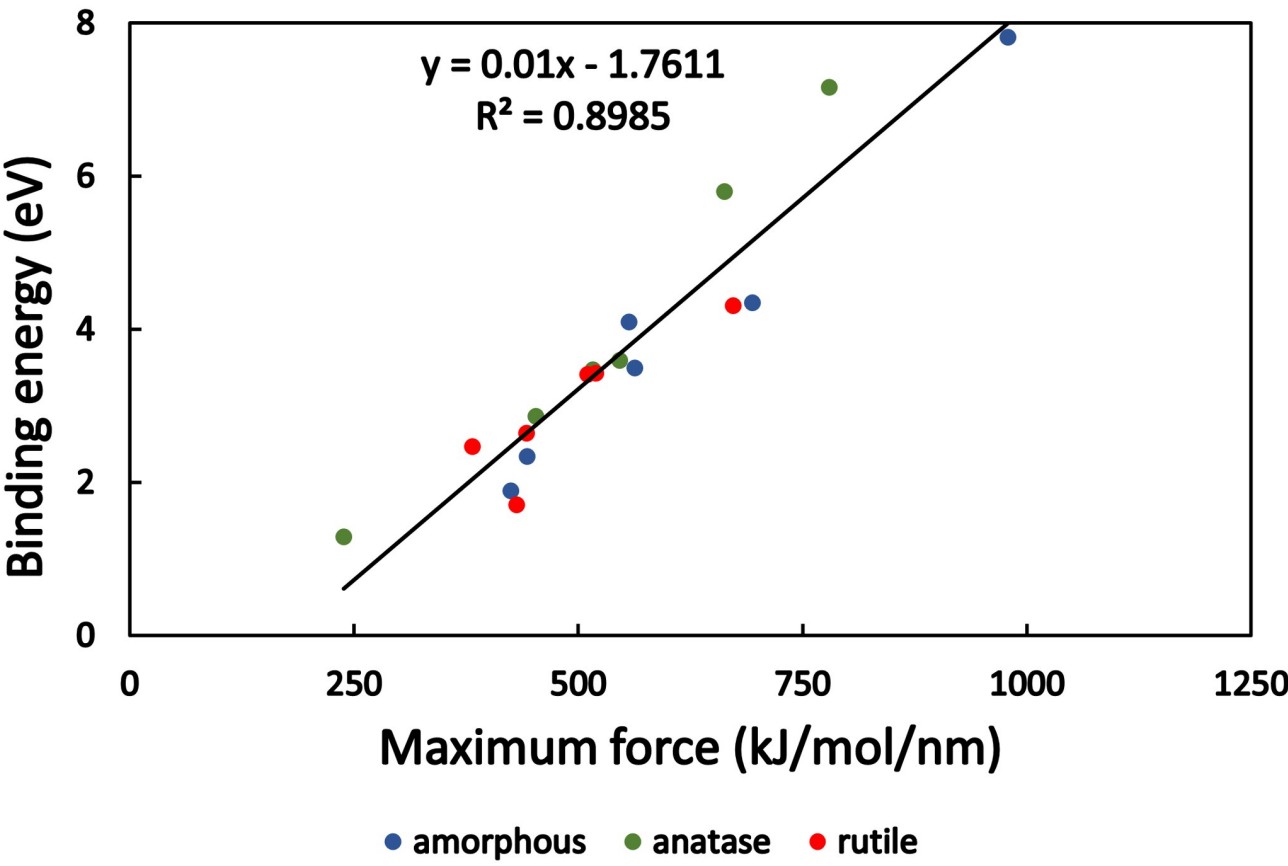

**Fig 8. Binding energy as a function of maximum force.** Both values were determined from the pulling simulation diagrams. The linear fit shows the correlation between these quantities.

the D (Asp) of RGD and LGD, as shown by Fig 6b and 6d, or the amine part of the N-terminal K (Lys) in case of the KRSR peptide. However, the ratio of the period that the peptides spend in the bonded form on the surface to the total simulation time varied considerably (Figs 4 and 5). This is related to the strength of the bonds formed between the investigated peptides and the TiO$_2$ surfaces. Distinct peptides may assume various conformations during adsorption to titanium dioxide surfaces. Frequently, they interact with the surface only via their N-terminal amino acid residue, or through the C-terminal one. It may also happen, however, that both the N-terminal and the C-terminal amino acid residue interact, simultaneously, with the TiO$_2$ surface (c.f. Fig A3 in Appendix). As most of the peptides detach from the surface during the 500 ns simulation time, they potentially adopt several different molecular conformations. The atoms in the simulations have a certain thermal energy (kinetic energy) resulting from the temperature, this may provide enough energy to break lower binding energies.

As shown in Fig A3 in Appendix the N terminal of the peptides (blue curves) can be closest to the oxygen atom of the TiO$_2$ surface with a low coordination number of up to ~0.15 nm. The adhering C terminal of the peptides (marked in red in Fig A3 in S1 Appendix) are located at a distance of ~0.3 nm. In this latter case, the binding is mediated through water, the atoms of the peptide are not directly bound to the TiO$_2$ surface but to the adsorbed H$_2$O molecules on it. This is a frequent adsorption type of the tested biomolecules in our simulations.

Our MD simulations show that the adhesion of tripeptides and tetrapeptides to the amorphous surface lasted the longest, the maximum tensile forces and binding energies were the highest on average, too. This may be due to the greater variety of binding sites to which peptides might bind after a certain diffusion-attachment-detachment series (see Fig 4). Liu *et al* performed MD simulations on amorphous TiO₂ as well, to study the interactions of TiO₂ nanoparticle surfaces with the twenty standard amino acids in an aqueous environment [35]. The amorphous TiO₂ nanoparticle was created with a 1.7 nm radius and, the adsorption probability (a similar quantity to our adsorption time ratio value, see Fig 5) was calculated by counting the number of simulation frames where side chain heavy atoms (C, N, O) approached the TiO₂ surface closer to 0.5 nm. Liu *et al.* found that compared to their non-polar and aromatic counterparts, polar amino acids, including the positively charged R (Arg) and K (Lys) as well as the negatively charged D (Asp) and Q (Glu), attached with a higher probability to the TiO₂ nanoparticle surface [35]. These results are consistent with our findings with the peptides tested. We observed that substituting the positively charged N-terminal K (Lys) with non-polar L (Leu) resulted in a decrease of the adsorption time ratio on all Ti surfaces (compare KRSR and LRSR, Fig 5a). Similarly, changing the positively charged N-terminal R (Arg) to non-polar L (Leu) decreased the adsorption time ratio of the relevant tripeptides on anatase and rutile surfaces during 500 ns simulations (compare RGD and LGD, Fig 5a).

The curves resulting from the pull simulation may be divided into three different sections (see Fig 7 and Fig A2 in S1 Appendix): in the first part, the peptide remained bonded to the TiO₂ surface, and the pulling force was linearly increasing with the distance (this is the elastic part of the pulling). After reaching a certain pulling-force threshold, however, the peptide leaves the TiO₂ surface. After this point—in the second part of the pulling—the force decreases as the distance to the surface is increasing. In the third part of the pulling, the molecules enter into the bulk water phase and the pulling force is oscillating around zero, as the water molecules also interact with the biomolecules even when they are detached from the surface.

In our study, the relationship between the maximum pulling forces and bond energies was examined as well, which showed a linear relationship as can be seen in Fig 8. The binding of a 13-residue-long biomolecule on an anatase TiO₂ surface was studied by Arcangeli *et al.* with pull MD simulations [36]. The center of mass (COM) of the biomolecule was pulled until its detachment from the TiO₂ surface. In their study, a higher maximum pulling force (~1500 kJ·mol⁻¹·nm⁻¹) was obtained for a longer polypeptide chain. However, this is consistent with our results (the maximum pulling force was ~1000 kJ·mol⁻¹·nm⁻¹, see Fig 8), as their molecule adsorbed to the surface via more than one charged amino acids. The study highlighted that mainly the charged R (Arg) amino acid constituent could diffuse through the adsorbed water layer and reach the surface by breaking the original H-bonds and forming new ones [36].

First principle calculations by Sowmiya *et al* showed that RGD adsorption energy on the TiO₂ surface ranged between 1.6 and 1.9 eV [37]. In their other study investigating P (Pro) and G (Gly) bound to anatase, the calculated dissociative adsorption energy values were 1.8 eV and 2.52 eV, respectively [38]. Compared to these values, our results show slightly higher binding energies for the peptides to TiO₂ surfaces. According to our data, changing the charged N-terminal amino acid to a non-charged one reduces the binding energies to all TiO₂ surfaces. For example, the change from KRSR to LRSR results in the binding energy change on anatase from 3.42 eV to 1.29 eV and rutile from 3.42 eV to 1.71 eV, while the change was not as pronounced in the case of amorphous TiO₂, from 4.35 eV to 4.1 eV.

Zhang *et al.* also studied the RGD tripeptide adsorption to TiO₂ surfaces via MD. The RGD behavior in vacuum and aqueous environment on the 110 rutile, 101 anatase, and a step-edged like surface was investigated [39]. If the peptide was standing on the N terminal the binding

energy was 43.99 kcal/mol (1.9 eV), standing on the O 37.77 kcal/mol (1.64 eV), while lying on the surface 39.71 kcal/mol (1.72 eV). Interestingly there is a higher difference in the case of anatase: standing on N 41.62 kcal/mol (1.8 eV), standing on O 28.52 kcal/mol (1.23 eV), and 23.8 kcal/mol (1.03 eV) lying on the surface. These energy values are similar to our results; however, the crystal structure of the surface was also an important factor that affected binding energies [39]. In addition, it was also shown that the surface step edges change the adsorption process drastically and the RGD peptide was less able to bind to the surface [39].

We would like to add that according to density functional theory (DFT) calculations, arginine interacts with anatase TiO₂ surface oxygen atoms via the protons of its guanidinium group [40]. DFT calculations were performed to characterize the binding of arginine to a nano-TiO₂ cluster as well [41], whereas Köppen et al. compared adsorption energies of arginine to anatase and rutile surfaces [42]. It is interesting to note, that in a simulation study glycine (G) and diglycine also interacted with a TiO₂ surface [43]. In case of the RGD tripeptide, however, a computational study indicated that the C-terminal aspartic acid (D) plays an important role in binding to the rutile TiO₂ (110) surface [44].

The initial process of cell adhesion to surfaces is very important for osseointegration and this can be assisted by peptide treatments of dental and bone implants. The potential clinical importance of our results lies in the fact that the MD simulations identified several tripeptides and tetrapeptides, which might be useful for planning further *in vitro* and *in vivo* experiments. As on real Ti implants, several crystalline structures can coexist, the averaging of adsorption time ratio and binding energy for the simulated three surface types in each biomolecule may approach a more realistic scenario. The mean values of the simulations for the six investigated molecules can be seen in Fig 9.

Regarding the average adsorption time ratio, the LRSR tetrapeptide has lower value than all the other molecules. The average binding energies show a broader variation. The lowest binding energies were obtained for the L (Leu) peptide-containing biomolecules (LGD and LRSR), while the highest average binding energies were obtained for the RSR-containing peptides (KRSR and RSR) and RGD. In summary, the K (Lys) and R (Arg) amino acid parts are dominantly bound to TiO₂ surfaces, while the non-charged L (Leu) peptide suppresses the adhesion.

Molecular dynamics simulations, as presented in this work, can be used to follow the binding of multiple peptide compositions on multiple types of surfaces, making it easier to select the appropriate peptide sequence with stable adhesion. The practical importance of our MD simulations is that surface type determines the adhesion less than the structure of the tri- or tetrapeptide molecule. Although the role in the adhesion of the amino and carboxyl groups of the end-terminal amino acid is important, according to our results, the adhesion is strongly influenced by the type of the peptide sequence.

The biointegration of dental implants can be modified by adsorbing polypeptides on their surfaces prior implantation. By tailoring peptide sequences bioactive or antibacterial surfaces can be developed [45]. The adhesion properties of each type of polypeptide can be investigated with MD simulations prior experimental studies. As a result, a reduced number of molecules and surface types (crystal structures) can be used in the experiments.

## 5. Conclusion

Adsorption of peptides and biomolecules on TiO₂ surfaces can affect the biocompatibility and functional properties of the Ti implants and can also influence the behavior of cells that come into contact with the surface. The aim of our research was to compare the adsorption of several peptides implicated in cell adhesion and tissue formation, and their variants, on different

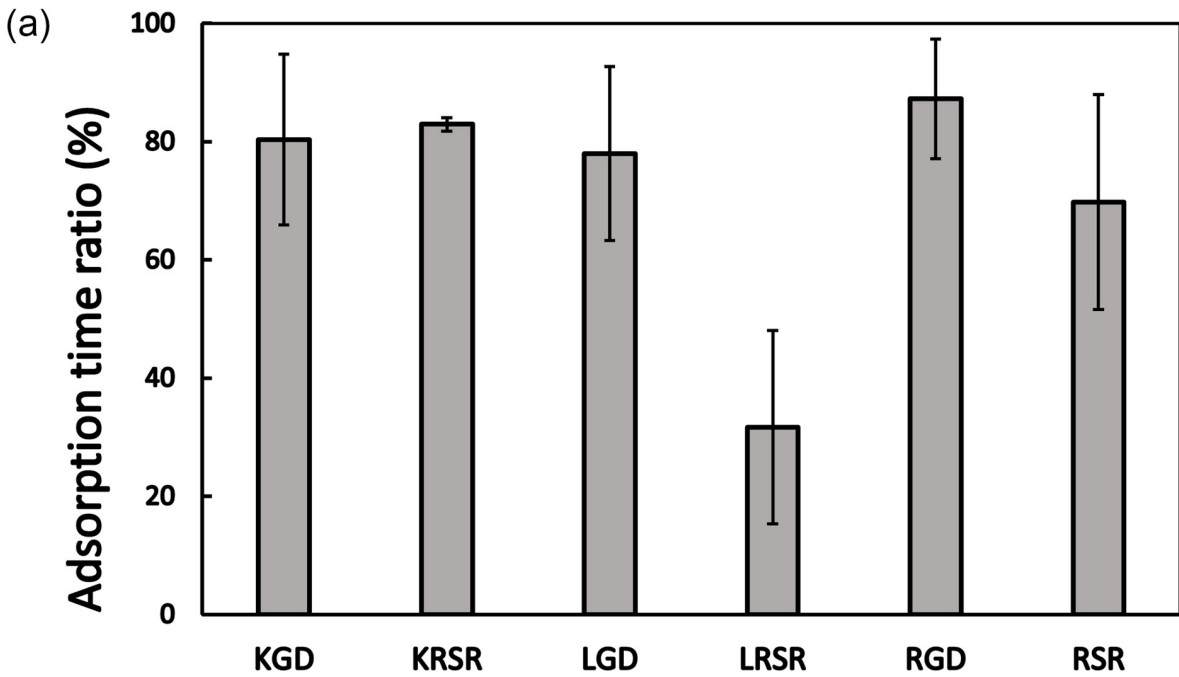

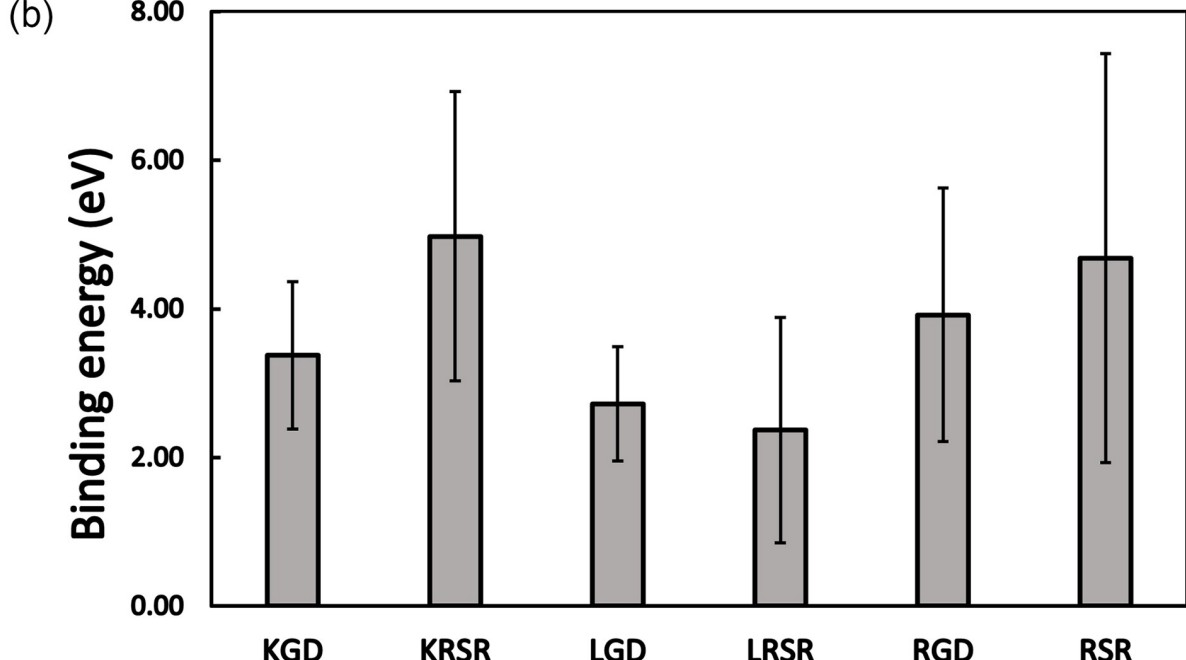

**Fig 9. (a) Adsorption time ratio and (b) binding energies averaged for the three TiO$_2$ surface types for each biomolecule with standard deviations.**

structures of TiO$_2$ surfaces. MD simulations were performed to model the adsorption processes of 6 different peptides to 3 different surface models of TiO$_2$. To compare the adsorption energies pulling MD simulations were performed and the binding energies from the force-distance curves were calculated.

The tripeptides and tetrapeptides migrated from the bulk water phase to the TiO$_2$ surface, where they formed a bond with the surface. The adhesion time ratio is the highest on the amorphous surface, which may be explained by the higher number of different forms of binding sites. All the peptides adhered to the amorphous TiO$_2$, rutile, and anatase surfaces except LRSR.

The binding energy and the maximum pulling force during the pull simulation showed a linear relationship. The calculated binding energies from the pull curves gave few eV values and made it possible to compare and quantify a difference between the adhesion to different TiO$_2$ surfaces for the different peptides. The KGD, RGD, and KRSR showed the highest adsorption time ratios and the KRSR had the highest average binding energy. In summary, our results are in agreement with the previous theoretical and experimental results on the adhesion of RGD and KRSR to TiO$_2$ and show that positively charged or dipole biomolecules may have a perspective in future applications to functionalize the surface of Ti implants.

Although we simulated the binding of six peptides to three kinds of TiO$_2$ surfaces in the present study, we think that further research is needed to establish the generalizability of the results. On the basis of the above conclusions the planning of further simulations, that serve future practical applications, are more straightforward.

## Supporting information

**S1 File. Peptides on TiO$_2$ surfaces videos.** Short videos representing the interaction of the 6 different biomolecules (KGD, KRSR, LGD, LRSR, RGD and RSR) with the 3 different TiO$_2$ surface (anatase, amorphous and rutile).
(DOCX)

**S2 File. LRSR trajectory on amorphous surface.** A representative trajectory file of the LRSR peptide and amorphous TiO$_2$ surface MD simulation without the ions and water molecules.
(DOCX)

**S3 File. Force-distance data.** Pull simulation force-distance curve data for the binding energy calculations.
(XLSX)

**S1 Appendix.**
(DOCX)

## Acknowledgments

We acknowledge KIFÜ for awarding us access to the HPC resources based in Hungary. Project ID: fimplant.

## Author Contributions

**Conceptualization:** Tamás Tarjányi, Ferenc Bogár, János Minárovits, Zsolt Tóth.

**Data curation:** Tamás Tarjányi.

**Funding acquisition:** János Minárovits.

**Investigation:** Tamás Tarjányi.

**Methodology:** Tamás Tarjányi.

**Supervision:** János Minárovits, Zsolt Tóth.

**Validation:** Ferenc Bogár, János Minárovits, Zsolt Tóth.

**Writing – original draft:** Tamás Tarjányi.

**Writing – review & editing:** Tamás Tarjányi, Ferenc Bogár, János Minárovits, Márió Gajdács, Zsolt Tóth.

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
