## [Decision Letter · Decision Letter 0]

27 Apr 2023

PONE-D-23-04372Interaction of biomolecules with anatase, rutile and amorphous TiO2 surfaces: A molecular dynamics studyPLOS ONE

Dear Dr. Tóth,

Thank you for submitting your manuscript to PLOS ONE. After careful consideration, we feel that it has merit but does not fully meet PLOS ONE’s publication criteria as it currently stands. Therefore, we invite you to submit a revised version of the manuscript that addresses the points raised during the review process.  Please submit your revised manuscript by Jun 11 2023 11:59PM. If you will need more time than this to complete your revisions, please reply to this message or contact the journal office at plosone@plos.org. When you're ready to submit your revision, log on to https://www.editorialmanager.com/pone/ and select the 'Submissions Needing Revision' folder to locate your manuscript file Please include the following items when submitting your revised manuscript:A rebuttal letter that responds to each point raised by the academic editor and reviewer(s). You should upload this letter as a separate file labeled 'Response to Reviewers'.A marked-up copy of your manuscript that highlights changes made to the original version. You should upload this as a separate file labeled 'Revised Manuscript with Track Changes'.An unmarked version of your revised paper without tracked changes. You should upload this as a separate file labeled 'Manuscript'.

We look forward to receiving your revised manuscript.

Kind regards,

Parag A. Deshpande

Academic Editor

PLOS ONE

Reviewers' comments:

Reviewer's Responses to Questions

**Comments to the Author**

1. Is the manuscript technically sound, and do the data support the conclusions?

Reviewer #1: Yes

Reviewer #2: Partly

2. Has the statistical analysis been performed appropriately and rigorously? 

Reviewer #1: Yes

Reviewer #2: Yes

3. Have the authors made all data underlying the findings in their manuscript fully available?

Reviewer #1: Yes

Reviewer #2: Yes

4. Is the manuscript presented in an intelligible fashion and written in standard English?

Reviewer #1: Yes

Reviewer #2: Yes

5. Review Comments to the Author

Reviewer #1: The research paper presented focuses on the adsorption of six biomolecules onto three different polymorphic structures of TiO2 through molecular dynamics (MD) simulations. The authors employed various analyses to evaluate the results, including distance variations between the biomolecules and the TiO2 surfaces, adsorption time ratios, force-distance diagrams, and binding energies.

Overall, the research paper is well-structured, and the methods employed are appropriate for the research question. The authors provide clear explanations of the MD simulations, and the results are presented in an organized and understandable manner. Additionally, the authors provide a comprehensive discussion of their findings, including the limitations of their study and potential future research directions.

One strength of the paper is the use of multiple techniques to analyze the adsorption of the biomolecules, which provides a more comprehensive understanding of the molecular interactions with the TiO2 surfaces. Furthermore, the authors present clear visualizations of their results, which helps to interpret the data.

However, the paper could benefit from more discussion on the practical implications of their findings. While the authors discuss the potential use of TiO2 surfaces in biomaterials, they do not provide clear examples of how their findings could be applied in the development of new materials or medical devices.

Moreover, the study has some limitations. The authors acknowledge that their simulations only capture a brief snapshot of the adsorption process and that the experimental conditions may differ from those of the simulations. Additionally, the study only considers six biomolecules, and the generalizability of the results to other molecules or surfaces is unclear.

Overall, the research paper provides valuable insights into the adsorption of biomolecules onto TiO2 surfaces. However, further research is necessary to determine the practical implications of these findings and to establish the generalizability of the results.

Reviewer #2: Reviewer Comments

Manuscript No: PONE-D-23-04372

Title: Interaction of biomolecules with anatase, rutile and amorphous TiO2 surfaces: A molecular dynamics study

In this study, the authors performed molecular dynamics simulations to understand the binding mechanisms and adhesion properties of six peptides such as tripeptide RGD (Arg-Gly-Asp), and its variants KGD (Lys-Gly-Asp) and LGD (Leu-Gly-Asp) as well as the tetrapeptides KRSR (Lys-Arg-Ser-Arg), its variant LRSR (Leu-Arg-Ser-Arg) and its truncated version RSR (Arg-Ser-Arg) over the anatase, rutile and amorphous TiO2 surfaces that can also facilitate cell adhesion and it impacts the osseointegration of peptide-coated implants. Authors calculated the binding energies of peptides adsorption over the TiO2 surfaces and found that LRSR and anatase and LRSL and rutile exhibit the smallest binding energies whereas the highest binding energies were observed for RSR and amorphous and KRSR and anatase. Also, they reported the residence time of peptides binding over the three surfaces of TiO2. Further, the authors performed the pulling simulations to characterize the nature of binding of various peptides on TiO2 surface.

The manuscript is probably publishable in PLOS ONE. However, the authors need to address some key questions carefully before it is considered for publication.

1) Fig 1: The geometries shown in Figure 1 a, b, and c representing anatase, rutile and amorphous TiO2 are not clear, and it is difficult to distinguish between the three phases of TiO2 from Figure 1. The authors should highlight the structural differences among the three phases of TiO2 clearly.

2) This question is related to the adsorption configurations of peptides over TiO2 surfaces. The authors explored different tripeptides and tetrapeptides and showed in Figure 6 that the RGD and LGD adsorbed over Ti atom of the TiO2 surface via the interaction of carboxylic groups. Did authors consider the interaction of other functional groups of the peptides such as amide/amine groups or C=O since the authors mentioned in the Discussions part of the manuscript that the binding of KRSR peptide occurs via the interaction of amine part of the N-terminal K(lysine)?

3) On the surface of TiO2, there are different active sites exist including Ti4c (four coordinate), Ti5c (five coordinated) and Ti6c (six coordinated) and oxygen O2c and O3c sites, respectively. Did the authors consider (or) check the site-specific adsorption of these peptides over the TiO2 surfaces? If yes, then please provide the details of which site is more active for the adsorption of these peptides in the manuscript!

4) In Figure 8, the authors reported a linear fit between Binding energy and maximum force. The binding energies and the maximum forces for all the 18 points (6 points over each surface) were represented with the same color which makes the reader difficult to identify which point is for what surface. It would be better if the authors color the points as per the surface the way they colored the bar charts in Figure 5 (green-anatase, red-rutile and blue-amorphous) for the better understanding and identification of BE and Max. forces based on the surfaces that the authors are referring to.

5) Did the authors see any diffusion of these peptides to the subsurface layers of amorphous TiO2 during the MD simulations? If yes, then how strongly these peptides bind with the subsurface layers of amorphous TiO2.

6) There are some computational papers on the adsorption of amino acids over TiO2 surfaces and nanoparticles that the authors should cite, (i) Koch R, Lipton A S, Filipek S, Renugopalakrishnan V, Arginine interactions with anatase TiO2 (100) surface and the perturbation of 49Ti NMR chemical shifts – a DFT investigation: relevance to Renu-Seeram bio solar cell. Journal of Molecular Model 2011, 17, 1467-1472

(ii) Koppen S, Bronkalla O, Langel W, Adsorption Configurations and Energies of Amino Acids on Anatase and Rutile Surfaces. Journal of Physical Chemistry C, 2008, 112, 13600-13606

(iii) Monti S, van Duin A C T, Kim S-Y, Barone V, Exploration for the Conformational and Reactive Dynamics of Glycine and Diglycine on TiO2: Computational Investigations in the Gas Phase and in Solution, Journal of Physical Chemistry C, 2012, 116, 5141-5150.

(iv) Muir J M R, Costa D, Idriss H, DFT computational study of the RGD peptide interaction with the rutile TiO2 (110) surface, Surface Science, 2014, 624, 8-14.

(v) Sai Phani Kumar V, Verma M, Deshpande P A, On interaction of arginine, cysteine and guanine with a nano-TiO2 cluster, Computational Biology and Chemistry, 2020, 86, 107236.

6. PLOS authors have the option to publish the peer review history of their article (what does this mean?). If published, this will include your full peer review and any attached files.

Reviewer #1: No

Reviewer #2: No

---

## [Author Response · Author response to Decision Letter 0]

30 Jun 2023

The Response to Reviewers file with images uploaded as a PDF file. Here the text of the response is copied.

Answers to the reviewers

Reviewer #1: The research paper presented focuses on the adsorption of six biomolecules onto three different polymorphic structures of TiO2 through molecular dynamics (MD) simulations. The authors employed various analyses to evaluate the results, including distance variations between the biomolecules and the TiO2 surfaces, adsorption time ratios, force-distance diagrams, and binding energies.

ANSWER: The authors thank the reviewer for the comments and the above summary of the manuscript. These were valuable for the authors to revise the manuscript. All observations were taken into account and the necessary corrections made.

Reviewer #1: Overall, the research paper is well-structured, and the methods employed are appropriate for the research question. The authors provide clear explanations of the MD simulations, and the results are presented in an organized and understandable manner. Additionally, the authors provide a comprehensive discussion of their findings, including the limitations of their study and potential future research directions.

ANSWER: The authors thank these positive comments, too. We took care to produce a manuscript that presents our findings and conclusions in a well-edited and reader-accessible format.

Reviewer #1: One strength of the paper is the use of multiple techniques to analyze the adsorption of the biomolecules, which provides a more comprehensive understanding of the molecular interactions with the TiO2 surfaces. Furthermore, the authors present clear visualizations of their results, which helps to interpret the data.

ANSWER: The authors appreciate these comments as well, especially because it seems to be that some of the approaches used in this work (e.g. determination of binding times and force-distance functions) has not been applied yet to the topic studied.

Reviewer #1: However, the paper could benefit from more discussion on the practical implications of their findings. While the authors discuss the potential use of TiO2 surfaces in biomaterials, they do not provide clear examples of how their findings could be applied in the development of new materials or medical devices.

ANSWER: The application possibilities of the results presented in the manuscript: 

Covering the surface of medical or dental implants by peptides or peptide sequences, one can change or improve their surface properties. First of all, the energetic states of the surfaces, and accordingly the hydrophilic or hydrophobic properties can be changed. As a result, the onset of osseointegration can be selectively determined, since when the implant is inserted into the bone, the osteoblast cells can only adhere if the bodily fluids wet the implant surface. The bioactivity or antibacterial effect of peptide or polypeptide coverings can be utilized, too, on the implant surfaces. 

According to the observation of the reviewer the discussion is extended with the following text (manuscript line numbers 374-378):

“The biointegration of dental implants can be modified by adsorbing polypeptides on their surfaces prior implantation. By tailoring peptide sequences bioactive or antibacterial surfaces can be developed [45]. The adhesion properties of each type of polypeptide can be investigated with MD simulations prior experimental studies. As a result, a reduced number of molecules and surface types (crystal structures) can be used in the experiments.” 

Reviewer #1: Moreover, the study has some limitations. The authors acknowledge that their simulations only capture a brief snapshot of the adsorption process and that the experimental conditions may differ from those of the simulations. Additionally, the study only considers six biomolecules, and the generalizability of the results to other molecules or surfaces is unclear.

ANSWER: The authors agree with the opinion of the reviewer: for a given application, it is important first to study the binding mechanism of peptide to the surface and determine the molecular adhesion mechanisms. In this work we investigated the adsorption properties of “only” six tri- and tetrapeptides on thermally relaxed TiO2 surfaces with two different, usual crystal structures rutile and anatase and an amorphous structure, too. Each interaction was calculated for 500 ns simulation time, which is long enough to investigate several attachment and detachment series. On this basis the adhering properties of other molecules can not yet be determined. However, one of the general results of this work is that the adsorption properties on the different surfaces do not differ significantly. As a consequence, the results of MD simulations made on a usual (e.g. rutile) surface can also be transferred to the mixed (mostly polycrystalline) crystal surfaces of the real (dental or bone) implants. 

The generalizability of the results to other molecules can be summarized as follows: peptide sequences have different adhesion dynamics, which in the case of 3-4 part peptide sequences is partly determined by the end-terminal amino acid and, according to our results, the adhesion is also influenced by the type of the peptide sequence. The stability, biocompatibility and bioactivity of implant surfaces, modified by peptides, still have to be verified by experiments. The practical applicability of supercomputer simulations, like ours, is given by the fact that molecular dynamics studies can have a significant contribution to reducing the number of experimental parameters, so that sequences that have already been qualified by MD simulations can be used in the experiments. Our results of our molecular dynamic studies, for example in the case of RGD and KRSR, are consistent with the experimental experience published in the literature; these sequences are able to form a more stable and permanent connection with the TiO2 surfaces. 

Our molecular dynamics simulations show the importance of surface radicals that govern physical and chemical bonding, which are more pronounced in the revised manuscript. On certain implant surfaces, these radicals allow the spontaneous and irreversible immobilization of peptides.

According to the observation of the reviewer the text of the discussion is extended with the following text (manuscript line numbers 368-373):

“Molecular dynamics simulations, as presented in this work, can be used to follow the binding of multiple peptide compositions on multiple types of surfaces, making it easier to select the appropriate peptide sequence with stable adhesion. The practical importance of our MD simulations is that surface type determines the adhesion less than the structure of the tri- or tetrapeptide molecule. Although the role in the adhesion of the amino and carboxyl groups of the end-terminal amino acid is important, according to our results, the adhesion is strongly influenced by the type of the peptide sequence. ”

Reviewer #1: Overall, the research paper provides valuable insights into the adsorption of biomolecules onto TiO2 surfaces. However, further research is necessary to determine the practical implications of these findings and to establish the generalizability of the results.

ANSWER: Thank you for the positive evaluation of our results on adsorption of biomolecules onto TiO2 surfaces. According to the question of the generalizability of the results, the conclusions are extended with the following text (manuscript line numbers 399-402):

“Although we simulated the binding of six peptides to three kinds of TiO2 surfaces in the present study, we think that further research is needed to establish the generalizability of the results. On the basis of the above conclusions the planning of further simulations, that serve future practical applications, are more straightforward.”

The authors express their appreciation to the reviewer for reading the manuscript and calling our attention to extend the discussion and conclusion part with the practical applications and generalizability issues of our results. In our opinion this helped to improve the quality of the manuscript. 

Reviewer #2: Reviewer Comments:

The authors thank the reviewer for the specific comments. These were valuable for the authors to consider the adhesion peculiarities of the studied surfaces and molecules. All observations were taken into account and answered to the best of our knowledge. New representations of the results of the simulations were produced and the corrections in the manuscript were performed.

Reviewer #2: Manuscript No: PONE-D-23-04372

Title: Interaction of biomolecules with anatase, rutile and amorphous TiO2 surfaces: A molecular dynamics study

In this study, the authors performed molecular dynamics simulations to understand the binding mechanisms and adhesion properties of six peptides such as tripeptide RGD (Arg-Gly-Asp), and its variants KGD (Lys-Gly-Asp) and LGD (Leu-Gly-Asp) as well as the tetrapeptides KRSR (Lys-Arg-Ser-Arg), its variant LRSR (Leu-Arg-Ser-Arg) and its truncated version RSR (Arg-Ser-Arg) over the anatase, rutile and amorphous TiO2 surfaces that can also facilitate cell adhesion and it impacts the osseointegration of peptide-coated implants. Authors calculated the binding energies of peptides adsorption over the TiO2 surfaces and found that LRSR and anatase and LRSL and rutile exhibit the smallest binding energies whereas the highest binding energies were observed for RSR and amorphous and KRSR and anatase. Also, they reported the residence time of peptides binding over the three surfaces of TiO2. Further, the authors performed the pulling simulations to characterize the nature of binding of various peptides on TiO2 surface.

The manuscript is probably publishable in PLOS ONE. However, the authors need to address some key questions carefully before it is considered for publication.

Reviewer #2: 1) Fig 1: The geometries shown in Figure 1 a, b, and c representing anatase, rutile and amorphous TiO2 are not clear, and it is difficult to distinguish between the three phases of TiO2 from Figure 1. The authors should highlight the structural differences among the three phases of TiO2 clearly.

ANSWER: The authors thank the reviewer for drawing the attention to more appropriate visualization of the crystal structures used in the simulations. We made new snapshots of the three different TiO2 forms focusing on the surface at a closer look to make the differences more visible between anatase, rutile and amorphous structures. Instead of a perspective view an orthographic view was applied. In case of the amorphous structure the Ti and O atoms are randomly distributed. ( revised manuscript Figure 1)

(IMAGE)

Oxygen and titanium atoms are shown in red and pink, respectively. The visualization was done by VMD in orthographic mode.

Reviewer #2: 2) This question is related to the adsorption configurations of peptides over TiO2 surfaces. The authors explored different tripeptides and tetrapeptides and showed in Figure 6 that the RGD and LGD adsorbed over Ti atom of the TiO2 surface via the interaction of carboxylic groups. Did authors consider the interaction of other functional groups of the peptides such as amide/amine groups or C=O since the authors mentioned in the Discussions part of the manuscript that the binding of KRSR peptide occurs via the interaction of amine part of the N-terminal K(lysine)?

ANSWER: At the creation stage the peptides were introduced into the simulation system by inserting them into the aqueous environment with a greater distance than 1 nm from the TiO2 surface. The interaction between the peptides and the surface changes over time dynamically and takes many possible molecular conformations. Our observations indicated that the peptides are mainly adsorbed to the TiO2 surfaces with the charged parts that can be found for example in the terminals that are the amine and carboxylic groups.

We performed a further evaluation of the simulations and evaluated the distance of the first and last amino acid residues of the peptides from the titanium surface over time. The following figure shows these results:

(This figure is inserted to the Appendix as Fig A3. The new figure caption is (manuscript line numbers 439-440):

“Fig. A3 Time dependence of the distances of the N terminals (blue curves) and C terminals (red curves) of the studied biomolecules from the three kinds of TiO2 surfaces.”

(IMAGE)

The blue curves indicate the distance between the titanium oxide surface and the amino terminal (N-terminal) amino acid residue of the peptide investigated. The corresponding position of the carboxy terminal (C-terminal) amino acid residues are indicated by red curves.

There are certain interactions, where a charged N-terminal amino acid residue binds to the TiO2 surface, take for example the RGD peptide at 80 ns on the amorphous TiO2 which can be seen on next images:

(IMAGE) 

The carboxyl groups can be also the binding site which is the case for example LGD on amorphous surface after 220 ns which can be seen in next images:

(IMAGE)

Also both sites can interact with the TiO2 at the same time which makes the peptides a bridge like adsorption, for example the RSR at 400 ns (amorphous substrate):

(IMAGE)

The visualization was done by VMD, the TiO2 surface atoms were drawn with the VDW styles (1.0 van der Waals size) and the peptide with CPK representation.

According to the request of the reviewer, we extended the Discussion by adding the following text (manuscript line numbers 281-286):

„Distinct peptides may assume various conformations during adsorption to titanium dioxide surfaces. Frequently, they interact with the surface only via their N-terminal amino acid residue, or through the C-terminal one. It may also happen, however, that both the N-terminal and the C-terminal amino acid residue interact, simultaneously, with the TiO2 surface (c.f. Fig. A3 in the Appendix). As most of the peptides detach from the surface during the 500 ns simulation time, they potentially adopt several different molecular conformations.”

Reviewer #2: 3) On the surface of TiO2, there are different active sites exist including Ti4c (four coordinate), Ti5c (five coordinated) and Ti6c (six coordinated) and oxygen O2c and O3c sites, respectively. Did the authors consider (or) check the site-specific adsorption of these peptides over the TiO2 surfaces? If yes, then please provide the details of which site is more active for the adsorption of these peptides in the manuscript!

ANSWER: The molecular dynamics simulations do not tell exactly which atom bonds to which one. The actual structure in any time is defined by the given potentials and the bond can be guessed by which atom is in which one in immediate vicinity. More precisely, the peptide is attached to one of the potential wells of the potential field described by the surface atoms (including adsorbed water molecules). Our interfaces are "relaxed". The single crystal surface structure would not be relevant in this case. The peptide gets attached to the surface randomly, as a consequence that specific binding site is not predefined. The diagrams included in our answer to Question 2 (see Appendix Figure A3) show that the N-terminal amine groups (blue curves) can be the closest to the "low coordination number" oxygen atoms of the TiO2 crystal (distance: up to 0.15 nm). In case of the C-terminal carboxyl groups, marked in red, the distances are around 0.3 nm. The latter bindings, at 0.3 nm, seem to be mediated by H2O. As requested, we indicated in the manuscript which site is more active for the adsorption of the peptides studied.

(manuscript line numbers 289-294)

“As shown in Appendix Fig. A3 the N terminal of the peptides (blue curves) can be closest to the oxygen atom of the TiO2 surface with a low coordination number of up to ~0.15 nm. The adhering C terminal of the peptides (marked in red in Appendix Figure A3) are located at a distance of ~0.3 nm. In this latter case, the binding is mediated through water, the atoms of the peptide are not directly bound to the TiO2 surface but to the adsorbed H2O molecules on it. This is a frequent adsorption type of the tested biomolecules in our simulations.”

From the MD simulations we have seen cases where the potential of the O atoms, protruding from the TiO2 surface due to thermal migration, appears as a stronger binding site, so the role of the low-coordinated O atoms could be observed.

Reviewer #2: 4) In Figure 8, the authors reported a linear fit between Binding energy and maximum force. The binding energies and the maximum forces for all the 18 points (6 points over each surface) were represented with the same color which makes the reader difficult to identify which point is for what surface. It would be better if the authors color the points as per the surface the way they colored the bar charts in Figure 5 (green-anatase, red-rutile and blue-amorphous) for the better understanding and identification of BE and Max. forces based on the surfaces that the authors are referring to.

ANSWER: Figure 8 was modified according to the suggestion of the reviewer.

Reviewer #2: 5) Did the authors see any diffusion of these peptides to the subsurface layers of amorphous TiO2 during the MD simulations? If yes, then how strongly these peptides bind with the subsurface layers of amorphous TiO2.

ANSWER: Although the distance between each peptide and the TiO2 surface was monitored throughout the simulation time, we did not observe any time frame in which any of the studied peptides diffused through the first layer of TiO2 surface atoms. We also looked through all the frames in VMD and there was no closer contact than the minimum 0.15 nm separation. At least that's what we could observe on the timescale of 500ns. In a much longer (simulation) time, it might be possible that parts of the amino acids diffuse into the subsurface layer during the competitive binding events with water molecules that already adhered to TiO2.

Reviewer #2: 6) There are some computational papers on the adsorption of amino acids over TiO2 surfaces and nanoparticles that the authors should cite, 

(i) Koch R, Lipton A S, Filipek S, Renugopalakrishnan V, Arginine interactions with anatase TiO2 (100) surface and the perturbation of 49Ti NMR chemical shifts – a DFT investigation: relevance to Renu-Seeram bio solar cell. Journal of Molecular Model 2011, 17, 1467-1472

(ii) Koppen S, Bronkalla O, Langel W, Adsorption Configurations and Energies of Amino Acids on Anatase and Rutile Surfaces. Journal of Physical Chemistry C, 2008, 112, 13600-13606

(iii) Monti S, van Duin A C T, Kim S-Y, Barone V, Exploration for the Conformational and Reactive Dynamics of Glycine and Diglycine on TiO2: Computational Investigations in the Gas Phase and in Solution, Journal of Physical Chemistry C, 2012, 116, 5141-5150.

(iv) Muir J M R, Costa D, Idriss H, DFT computational study of the RGD peptide interaction with the rutile TiO2 (110) surface, Surface Science, 2014, 624, 8-14.

(v) Sai Phani Kumar V, Verma M, Deshpande P A, On interaction of arginine, cysteine and guanine with a nano-TiO2 cluster, Computational Biology and Chemistry, 2020, 86, 107236.

ANSWER: The authors are thankful for the reference to the literature mentioned in the reviewer's comments. We have read and studied the articles. Accordingly, the following paragraph was added to the discussion (manuscript line numbers 344-350):

“We would like to add that according to density functional theory (DFT) calculations, arginine interacts with anatase TiO2 surface oxygen atoms via the protons of its guanidinium group [40] (Koch et al., 2011). DFT calculations were performed to characterize the binding of arginine to a nano-TiO2 cluster as well [41] (Sai Phani Kumar et al., 2020), whereas Koppen et al. compared adsorption energies of arginine to anatase and rutile surfaces [42] (Koppen et al., 2008). It is interesting to note, that in a simulation study glycine (G) and diglycine also interacted with a TiO2 surface [43] (Monti et al., 2012). In case of the RGD tripeptide, however, a computational study indicated that the C-terminal aspartic acid (D) plays an important role in binding to the rutile TiO2 (110) surface [44]. (Muir et al., 2014).”

The authors are grateful to the reviewer for reading the manuscript and for the reviewer's comments. We believe that the technical content of the manuscript has been improved during the revision.

---

## [Decision Letter · Decision Letter 1]

20 Jul 2023

Interaction of biomolecules with anatase, rutile and amorphous TiO2 surfaces: A molecular dynamics study

PONE-D-23-04372R1

Dear Dr. Tóth,

We’re pleased to inform you that your manuscript has been judged scientifically suitable for publication and will be formally accepted for publication once it meets all outstanding technical requirements.

Kind regards,

Parag A. Deshpande

Academic Editor

PLOS ONE

Additional Editor Comments (optional):

Reviewers' comments:

Reviewer's Responses to Questions

**Comments to the Author**

1. If the authors have adequately addressed your comments raised in a previous round of review and you feel that this manuscript is now acceptable for publication, you may indicate that here to bypass the “Comments to the Author” section, enter your conflict of interest statement in the “Confidential to Editor” section, and submit your "Accept" recommendation.

Reviewer #1: All comments have been addressed

Reviewer #2: All comments have been addressed

2. Is the manuscript technically sound, and do the data support the conclusions?

Reviewer #1: Yes

Reviewer #2: Yes

3. Has the statistical analysis been performed appropriately and rigorously? 

Reviewer #1: N/A

Reviewer #2: Yes

4. Have the authors made all data underlying the findings in their manuscript fully available?

Reviewer #1: Yes

Reviewer #2: Yes

5. Is the manuscript presented in an intelligible fashion and written in standard English?

Reviewer #1: Yes

Reviewer #2: Yes

6. Review Comments to the Author

Reviewer #1: no further comments, all issue has been addressed by the authors in the last round of submission. Therefore I recommend publication

Reviewer #2: The authors addressed the reviewer's comments appropriately in the revised version of the manuscript and I recommend this article for publishing to PLOS One with this revised format.

7. PLOS authors have the option to publish the peer review history of their article (what does this mean?). If published, this will include your full peer review and any attached files.

Reviewer #1: No

Reviewer #2: No

---

## [Editor Report · Acceptance letter]

25 Aug 2023

PONE-D-23-04372R1 

Interaction of biomolecules with anatase, rutile and amorphous TiO2 surfaces: A molecular dynamics study 

Dear Dr. Tóth:

I'm pleased to inform you that your manuscript has been deemed suitable for publication in PLOS ONE. Congratulations! Your manuscript is now with our production department. 

Kind regards, 

on behalf of

Dr. Parag A. Deshpande 

Academic Editor

PLOS ONE